# Prediction of African Swine Fever Virus Inhibitors by Molecular Docking-Driven Machine Learning Models

**DOI:** 10.3390/molecules26123592

**Published:** 2021-06-11

**Authors:** Jiwon Choi, Jun Seop Yun, Hyeeun Song, Yong-Keol Shin, Young-Hoon Kang, Palinda Ruvan Munashingha, Jeongyeon Yoon, Nam Hee Kim, Hyun Sil Kim, Jong In Yook, Dongseob Tark, Yun-Sook Lim, Soon B. Hwang

**Affiliations:** 1Department of Oral Pathology, Oral Cancer Research Institute, Yonsei University College of Dentistry, Seoul 03722, Korea; YJS8714@yuhs.ac (J.S.Y.); 0212HEA@yuhs.ac (H.S.); MIGO77@yuhs.ac (N.H.K.); KHS@yuhs.ac (H.S.K.); JIYOOK@yuhs.ac (J.I.Y.); 2Met Life Sciences Co. Ltd., Seoul 03722, Korea; 3Enzynomics Co. Ltd., Yuseong-gu, Daejeon 34050, Korea; ykshin@enzynomics.com (Y.-K.S.); yhkang@enzynomics.com (Y.-H.K.); mpruvan@enzynomics.com (P.R.M.); jyyoon@enzynomics.com (J.Y.); 4Laboratory for Infectious Disease Prevention, Korea Zoonosis Research Institute, Jeonbuk National University, Iksan 54596, Korea; tarkds@jbnu.ac.kr; 5Laboratory of RNA Viral Diseases, Korea Zoonosis Research Institute, Jeonbuk National University, Iksan 54596, Korea; sbhwang@jbnu.ac.kr; 6Ilsong Institute of Life Science, Hallym University, Seoul 03722, Korea

**Keywords:** African swine fever virus, antiviral, molecular docking, machine learning

## Abstract

African swine fever virus (ASFV) causes a highly contagious and severe hemorrhagic viral disease with high mortality in domestic pigs of all ages. Although the virus is harmless to humans, the ongoing ASFV epidemic could have severe economic consequences for global food security. Recent studies have found a few antiviral agents that can inhibit ASFV infections. However, currently, there are no vaccines or antiviral drugs. Hence, there is an urgent need to identify new drugs to treat ASFV. Based on the structural information data on the targets of ASFV, we used molecular docking and machine learning models to identify novel antiviral agents. We confirmed that compounds with high affinity present in the region of interest belonged to subsets in the chemical space using principal component analysis and *k*-means clustering in molecular docking studies of FDA-approved drugs. These methods predicted pentagastrin as a potential antiviral drug against ASFVs. Finally, it was also observed that the compound had an inhibitory effect on *Asfv*PolX activity. Results from the present study suggest that molecular docking and machine learning models can play an important role in identifying potential antiviral drugs against ASFVs.

## 1. Introduction

African swine fever (ASFV) is an enveloped and double-stranded DNA virus belonging to the *Asfarviridae* family (genus *Asfivirus*), which contains a genome measuring 170–193 kbp, and replicates predominantly in the cytoplasm of macrophages [1]. The virus has a complex structure and a genome that mainly replicates in the cytoplasm of infected cells [2]. It has resulted in high mortality rates in pigs and is responsible for serious economic and production losses worldwide [3,4]. Recently, ASFV outbreaks reported in EU countries and Transcaucasia, especially in East Asian countries, have resulted in the culling of over one million pigs [5,6,7]. Vaccines and antiviral drugs can stop the spread of infection, but efforts to develop effective vaccines against ASFV have failed [8]. Therefore, there is a need for novel effective drugs and vaccines against ASFV. Currently, no vaccine or treatment is available against ASFV infection. Recent attempts to generate DNA vaccines, vectored vaccines, or live attenuated vaccines by defined gene deletions have yielded encouraging data through in vitro assays [9,10,11,12,13]. However, there are still major challenges with respect to the safety of live-attenuated ASFV vaccines, and their implementation is highly limited. Several studies have attempted to identify multiple compounds that can inhibit ASFV infections. Antiviral drugs with identified targets and known mechanisms can be classified into five distinct categories as follows: nucleoside analogs, interferons, plant-derived compounds, antibiotics, small interfering RNA, and CRISPR/Cas9 [14,15,16,17,18,19,20,21,22,23]. Other antiviral drugs such as apigenin, resveratrol, and oxyresveratrol show potent, dose-dependent anti-ASFV activity in vitro, but they have unknown targets and mechanisms. To date, there have been no effective drugs for clinical trials against ASFV, and there is an urgent need to develop preventive and therapeutic reagents.

ASFV is primarily replicated in the cytoplasm of swine macrophages, which is a major source of reactive oxygen species that cause constant damage to the ASFV genome [24,25,26,27]. To overcome potential DNA damage, such as apurinic/apyrimidinic (AP) sites and/or single strand breaks, the virus has evolved its own DNA repair system, including AP endonuclease (*Asfv*AP), a repair DNA polymerase (*Asfv*PolX), and a DNA ligase (*Asfv*LIG) [28,29,30]. Unlike the homologous proteins, the fidelities of both *Asfv*PolX and *Asfv*LIG are very low. Therefore, in addition to their critical role in genome stability maintenance, these enzymes play an important role in the strategic mutagenesis and genotype formation of ASFV [28,31,32,33]. Owing to their functional importance, ASFV repair enzymes have been extensively studied [34,35].

In this study, we focused on an enzyme that helps synthesize DNA, the ASFV DNA polymerase X or *Asfv*PolX, to identify novel targets for the prevention and treatment of ASFV. *Asfv*PolX plays a role in the DNA repair process of the ASFV genome, and viral replication is partially dependent on the function of *Asfv*PolX. In addition, *Asfv*PolX has several unique structural features, including a 5′-phosphate (5′-P) binding pocket, a His115-Arg127 platform, and hydrophobic residues Val120 and Leu123, which can all affect the catalytic efficiency of *Asfv*PolX. It can be assumed that blocking the binding pocket with therapeutics may inhibit *Asfv*PolX activity and thereby impair the DNA repair process of the viral genome [36]. In the present study, we used the crystal structure of two different types of *Asfv*PolX protein structures (*Asfv*PolX apo and *Asfv*PolX/DNA structures) to identify potential antiviral drugs using a novel integrated strategy that combines molecular docking and machine learning models [14,37]. Artificial intelligence and machine learning have been successfully applied in bioinformatics [38,39,40,41,42]. To identify novel antiviral inhibitors, we first conducted molecular docking studies of FDA-approved drugs based on the structural information of the targets of ASFV and predicted ten top-ranked compounds with high docking scores using principal component analysis (PCA) and *k*-means clustering. Pentagastrin, which had the highest binding affinity in both the apo and DNA complex structures of *Asfv*PolX, was chosen as an antiviral candidate drug. We also identified the dose-dependent inhibitory effect of pentagastrin on *Asfv*PolX polymerase activity.

The use of this combination strategy can provide an alternative approach to identifying potential antiviral drugs against ASFVs from molecules that have already been tested or approved for other diseases. Furthermore, the region of interest that only appears in groups with high binding affinity to ASFV targets identified based on eight molecular descriptors used in this study can provide key information regarding the unique chemical characteristics of antiviral drugs against ASFVs. These results demonstrate that active compounds with regions of interest in the chemical space may be expected to provide insights into potential ASFV inhibitors.

## 2. Results and Discussion

### 2.1. Identification of Ligandable Pockets

The putative binding sites of the *Asfv*PolX protein were identified using SiteMap, with an applied force field of OPLS_2005 in the Schrödinger suite. SiteMap uses the hydrophobicity and accessibility of a detected binding site to assess the likelihood of a small-molecule inhibitor binding to it. We denoted the four top-ranking surface clefts identified by the SiteMap module on the structure of *Asfv*PolX protein in complex with DNA (PDB ID: 5HRB) from site 1 to site 4 (Figure 1). Sites 1 and 3 were clearly distinct from the DNA binding sites; however, sites 2 and 4 were very close in the 5′-P binding pocket of *Asfv*PolX. We finally decided on sites 2 and 4 as the potential binding pockets of *Asfv*PolX to find antiviral drugs through docking simulations.

### 2.2. Docking Simulation to DNA Binding Site

To discover new antiviral drugs capable of inhibiting the activity of *Asfv*PolX, we first performed molecular docking using binding pockets identified through SiteMap. The preparation of datasets for molecular docking was performed with two different types of *Asfv*PolX protein structures (*Asfv*PolX apo and *Asfv*PolX/DNA structures), using FDA-approved drugs from the DrugBank database. The reported eight *Asfv*PolX complex structures demonstrated that *Asfv*PolX has one unique 5′-P binding pocket that can inhibit the activity of *Asfv*PolX and disrupt the DNA repair process of the ASFV genome [36]. Previous binding site analysis studies using SiteMap also identified the 5′-phosphate (5′-P) binding pocket of *Asfv*PolX as a potential binding pocket for antiviral inhibitors. To determine which conformations of the apo and DNA complex structures are structurally proximal conformations as template structures for virtual screening of antiviral compounds with high binding affinity, we implemented a molecular docking program using both the crystal structure of apo (PDB ID:1JAJ) and DNA complex structure (PDB ID:5HRB). We obtained 1729 drugs from the DrugBank database. After performing molecular docking, the 1JAJ and 5HRB datasets consisted of 1714 and 1712 compounds. Figure 2A,B show the histogram and boxplot distribution of the docking scores for the two structures. The distribution of docking score for the DNA complex structure was shifted to the left relative to that of the apo structure (Figure 2A). In addition, the mean docking scores of compounds in apo and DNA complex structures were −3.703 and −4.323, respectively, which implied that compounds for DNA complex structure could be predicted to have higher binding affinity than compounds for apo structure (Figure 2B). These results indicated that the *Asfv*PolX-DNA complex structure is a suitable template for determining compounds with high binding affinity and can also provide important guidance for the design and discovery of potential antiviral inhibitors for *Asfv*PolX.

### 2.3. Identification of New Chemical Space against AsfvPolX Inhibitor

To investigate the distinct features of molecules based on docking results for apo and DNA complex structures, eight molecular descriptors of each molecule were calculated and applied to PCA (Figure 3). The 1JAJ and 5HRB datasets consisted of 1714 and 1712 compounds based on the docking score. We then constructed the 1JAJ and 5HRB datasets which consisted of 675 and 1035 compounds, respectively, with binding free energy ≤−4.0 kcal/mol based on docking score. These datasets were selected for further PCA in this step. We also constructed a duplicated dataset consisting of 601 compounds with binding free energy ≤−4.0 kcal/mol for both apo and DNA complex structures. In PCA visualizations of all the three datasets with high binding affinity, we determined that the datasets were generally well distributed throughout the chemical space, whereas the three datasets with binding affinities ≥−4.0 kcal/mol were significantly concentrated in the center (Figure 3). Of the three datasets, 5HRB and duplicated datasets generated a lot of separately distributed points in the chemical space (Figure 3A). All the three datasets of compounds with low binding affinity show that the most densely packed regions were in the center (Figure 3B). PCA visualization of all three datasets containing compounds with binding affinity ≤−4.0 kcal/mol or ≥−4.0 kcal/mol (represented by “Non-binding” in Appendix A) shows that centralized distribution was clearly found in the non-binding dataset. From these results, we determined the region of interest present in 5HRB and duplicated the dataset with binding affinities ≤−4.0 kcal/mol as a new chemical space for finding a potential inhibitor against ASFVs, which was then preferentially chosen to select the final compound for experimental testing.

### 2.4. Molecular Descriptor-Based Clustering

To investigate the region of interest in the chemical space, we used *k*-means clustering for three datasets consisting of compounds with binding affinities ≤−4.0 kcal/mol based on eight molecular descriptors (Figure 4A). We expected that the diverse regions of 5HRB and duplicated dataset through PCA could provide several important chemical regions for finding compounds with higher binding affinities in the DNA-binding sites of ASFVs. For the datasets, we identified the optimal clustering consisting of groups of six clusters. These results indicated that physicochemical descriptor-based clustering can accurately provide a unique region for 5HRB and duplicated dataset containing compounds with high affinity, demonstrating that compounds with the best docking score in both ASFV protein structures are clustered together in the chemical space. For compounds in Clusters 1, 3, and 4, the mean docking scores were −5.27, −4.63, and −4.87, respectively (Figure 4B). The mean docking scores of compounds in Clusters 2, 5, and 6 were −6.0, −5.26, and −5.44, respectively, which implied that compounds of Cluster 2 could be predicted to have higher binding affinity than other clusters. This showed chemical diversity and regions of interest in chemical space. We showed that PCA-based clustering on the eight molecular descriptors used in this study could discriminate compounds with high binding affinity and provide valuable guidance on the design and discovery of potent antiviral drugs available for the prevention or treatment of ASFV infections.

### 2.5. Pentagastrin on AsfvPolX Polymerase Activity

Among the top-ranked ten ligands in Cluster 2, pentagastrin with the highest binding affinity in both apo and complex structures was chosen as a promising drug candidate. The list of the top-ranked ten ligands in Cluster 2 can be found in Appendix A. To measure the effect of pentagastrin on *Asfv*PolX activity, BL21 (DE3)-RIL cells carrying pET28-*Asfv*PolX plasmid were cultured in the presence of isopropyl β-d-1-thiogalactopyranoside, and then the *Asfv*PolX protein was purified using a Ni^2+^-nitrilotriacetic acid (Ni-NTA) column (Figure 5A). *Asfv*PolX protein was further purified using a glycerol gradient (Figure 5B). We first verified *Asfv*PolX polymerase activity. As shown in Figure 5C, *Asfv*PolX activity rapidly increased after the incorporation of nucleotides up to 200 ng of enzyme, and the activity gradually decreased thereafter. We then determined the effect of pentagastrin on *Asfv*PolX polymerase activity. As shown in Figure 5D, *Asfv*PolX activity was significantly decreased by pentagastrin in a dose-dependent manner. We showed that 100 µM of pentagastrin inhibited *Asfv*PolX activity by 60% (Figure 5D), indicating that pentagastrin may be a potent inhibitor of the ASFVs.

### 2.6. Binding Interaction Patterns of Pentagastrin

The binding interactions between pentagastrin and AsfvPolX protein, identified using Glide (Schrödinger, LLC, New York, NY, USA, 2020), showed that pentagastrin was fitted into a groove at the same DNA binding pocket, and that it had a similar conformation as that of 1nt-gap DNA1 (Figure 6A). In the docking model of pentagastrin, the phenyl rings docked into a hydrophobic cavity formed by residues Val120, Leu123, and Ile124 of *Asfv*PolX protein. The hydroxyl group of pentagastrin formed hydrogen bonds with Lys85 and Lys136 at the binding site (Figure 6B). The prediction of the binding poses of the other nine candidate ligands through molecular docking simulations is represented in Appendix A. Docking studies demonstrated that these residues contribute to the binding interaction of pentagastrin with the *Asfv*PolX protein and could be important for the design of new ligands.

## 3. Materials and Methods

### 3.1. Binding Site Analysis

The binding site was characterized using the SiteMap module of the Schrödinger software package (Schrödinger, New York, NY, USA using the high-resolution crystal structure, *Asfv*PolX, in complex with DNA (Protein Data Bank [PDB] ID, 5HRB) [43]. The SiteMap module can be used to identify and clarify potential binding sites (site points) that are most likely contribute to tight protein–ligand or protein–protein binding. The following physicochemical properties of the sites were calculated using the SiteMap program: size, volume, degree of enclosure/exposure, degree of contact, hydrophobic/hydrophilic character, hydrophobic/hydrophilic balance, and hydrogen-bonding possibilities (acceptors/donors). The algorithm of SiteMap determines the likeliness of a site point whose contribution depends on how close the site points are to the protein surface and how site points are sheltered from the solvent. These site points, determined by SiteMap, can be used to define the active site of a protein.

### 3.2. Docking Simulation

To examine the binding interactions of the *Asfv*PolX-ligand complexes, molecular docking studies were performed using Glide software (Schrödinger, New York, NY, USA), which uses an optimized potential for liquid simulations (OPLS)-2005 force field, and refinement was carried out according to the recommendations of the Schrödinger Protein Preparation Wizard. The ligand database consists of 2413 FDA-approved drugs obtained from DrugBank. Prior to screening, the database was filtered to select compounds with molecular weights >200 Da and 1729 compounds were obtained. LigPrep (Schrödinger, New York, NY, USA) was used to generate three-dimensional ligand structures. The active grid was generated using the Receptor grid application in the Glide module. On a defined receptor grid, flexible docking was performed using the standard precision mode of Glide [44,45]. The datasets of apo (PDB ID:1JAJ) and DNA complex structures (PDB ID:5HRB) contained 1714 and 1712 compounds, respectively, based on the Glide score.

### 3.3. Dataset Preparation

The dataset was generated from the DrugBank database (https://go.drugbank.com (DrugBank Release Version 5.1.4, 2 July 2019)), consisting of more than 11,000 drugs. Drugs in this database are classified into different types as follows: trial stage drugs, approved drugs, withdrawn drugs, and investigational drugs. In this study, we focused only on FDA-approved drugs (around 2413 compounds) (DrugBank Release Version 5.1.4, 2 July 2019). These compounds were downloaded in the form of a structure data file from the DrugBank database. We removed compounds with a molecular weight less than approximately 200, and thus obtained a dataset comprising 1729 drugs. To perform molecular docking, we used apo and DNA complex structures of the *Asfv*PolX target to obtain sets of apo (PDB ID:1JAJ) and DNA complex structures (PDB ID:5HRB), which consisted of 1714 and 1712 compounds, respectively, based on the docking score. Finally, 1JAJ and 5HRB datasets consisting of 675 and 1035 compounds with binding affinities ≤−4.0 kcal/mol based on docking score were subjected to further PCA analysis.

### 3.4. Principal Component Analysis and Cluster Analysis

A molecular descriptor is a numerical description that represents the physical and chemical information of a compound. The descriptor parameters for the datasets were generated using the molecular descriptor calculator included in the QikProp module of Schrödinger (Maestro, Schrödinger, LLC, New York, NY, USA, 2020). The calculated descriptors included molecular weight, number of hydrogen bond acceptors, number of hydrogen bond donors, ALogP, number of rotatable bonds, number of aromatic rings, number of rings, and polar surface area. The frequency distribution of physicochemical properties for the datasets is shown in Appendix A. To investigate and compare the distinct features of molecules based on the docking results for apo and DNA complex structures, the eight aforementioned molecular descriptors of each molecule were calculated. PCA is a multivariate statistical method used for exploratory data analysis. It allows the representation of the property space by projection into the principal component plane (PC1, PC2), which is the mathematical equivalent of taking a picture from the most favorable angle [46,47,48]. To extract the most important information from the dataset, PCA was employed to explore the chemical space of antiviral compounds as a function of the eight molecular descriptors, using the FactoMineR package in R [49,50]. PCA clustering was calculated for each dataset using the *k*-means clustering method within the Factoextra R package. All histograms and scatter plots were generated using the R software.

### 3.5. Plasmid Construction

To confirm the in vitro antiviral activity for the candidate compounds identified in this study, the full-length PolX gene (*O174L*) was manually synthesized (BIONICS) and cloned into the BamHI and HindIII sites of the pET28 vector.

### 3.6. Purification of PolX Proteins

The pET28-PolX plasmid was transformed into *Escherichia coli* BL21 (DE3)-RIL, and cells were grown at 37 °C until an A_600_ value of 0.5 was reached. Cells were then induced with 1 mM isopropyl β-d-1-thiogalactopyranoside for 3 h and collected by centrifugation. The cell pellet was resuspended in buffer T_300_ (50 mM Tris-HCl [pH 7.5], 300 mM NaCl, 0.1% Nonidet P-40, 10% glycerol, and 0.1 mM phenylmethylsulfonyl fluoride). The subscript number in T_300_ indicates the concentration of NaCl (mM). Cells were then lysed by sonication, and the cell extract was centrifuged for 30 min at 13,000 rpm. The supernatant was loaded onto a Ni-NTA column (Qiagen, Valencia, CA, USA) column and washed successively with low concentrations of imidazole (20 mM and 60 mM) in T_300_ until no protein was eluted, and the protein was eluted with 250 mM imidazole in the same buffer. The peak protein fractions were pooled and diluted in 50 mM NaCl. The pooled protein was then loaded onto a HiTrap SP HP column (GE Healthcare, Chicago, IL, USA) that was eluted with a linear gradient (50–600 mM) of NaCl in buffer T. The eluted fractions containing the highest concentration of protein were collected (eluting between 350 and 450 mM of NaCl), and an aliquot was subjected to glycerol gradient centrifugation (15–35% glycerol) at 54,000 rpm for 26 h in a Beckman SW55Ti rotor. Gradient fractions containing active PolX were stored at −70 °C until use.

### 3.7. Measurement of Polymerase Activity

Polymerase activity of PolX was measured using the polymerase activity assay kit (Enzynomics, Daejeon, Korea), according to the manufacturer’s instructions, with minor modifications. Briefly, PolX protein was mixed with pentagastrin (Sigma-Aldrich, St Louis, MO, USA) for 10 min at 4 °C and then further mixed with 2× reaction buffer and primed substrate. For PolX induction, the mixture was incubated for 30 min at 37 °C. Following inactivation of polymerase activity, the synthesized DNA was denatured at 95 °C for 20 min. For annealing DNA, the temperature was decreased at 65 °C for 10 min, 45 °C for 5 min, and 25 °C for 5 min. One hundred cycles of 5 s, with 0.5 °C temperature increments from 45 °C to 95 °C, were used for the melting curves to measure synthesized DNA contents using a CFX Connect real-time system (Bio-Rad Laboratories). The 2× reaction mixture was incubated with a standard substrate (S1, S2, S3) to create a standard curve.

## 4. Conclusions

In this study, we applied molecular docking and machine learning to build models from publicly available screening data (DrugBank) to predict antiviral drugs that could potentially inhibit ASFV. To determine which conformations of the apo and DNA complex structures are structurally proximal conformations as template structures for virtual screening, we first conducted a docking simulation using both structures. To investigate and compare molecules based on the docking scores for apo and DNA complex structures, eight molecular descriptors of each molecule were calculated. From molecular descriptor-based PCA, we determined that 5HRB and a duplicated dataset containing compounds with high binding affinity were generally well distributed throughout the chemical space. Conversely, the three datasets of compounds with low binding affinity were distributed concentrically around the center of the chemical space. In this work, the *k*-means clustering method was used to establish classification models for predicting whether compounds with high binding affinity are inhibitors of ASFVs. For the datasets, we identified the optimal clustering consisting of six clusters. In particular, compounds in Cluster 2 were found to have a much higher binding affinity on average compared to the other clusters. Therefore, we selected the top-ranked (with the top score) pentagastrin on both apo and DNA complex structures for further in vitro studies. Pentagastrin inhibited *Asfv*PolX activity by 60% at 100 μM concentrations, indicating that the strategy used in this study can reduce the processing time and cost required to identify potential antiviral drugs against ASFVs from molecules already tested or approved for other diseases. These studies to identify drugs with antiviral activity against ASFV have helped in determining active antiviral drugs available for the treatment of ASFV infections. The present study, performed through molecular docking and machine learning models, can play an important role in identifying potential antiviral drugs against ASFVs.

## Figures and Tables

**Figure 1 molecules-26-03592-f001:**
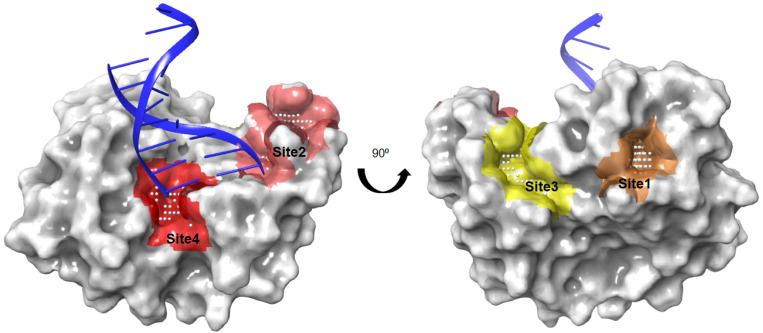
The four top-ranking surface pockets observed in the *Asfv*PolX protein identified by the SiteMap module of Schrödinger. Colored spaces represent the SiteMap predictions for four top-ranking surface clefts on the *Asfv*PolX protein (PDB ID: 5HRB, in grey surface representation): site 1: orange, 2: pink, 3: yellow, 4: red. The red and pink spaces correspond to the DNA binding site. Grey dots represent the cavity of the putative active site of the protein.

**Figure 2 molecules-26-03592-f002:**
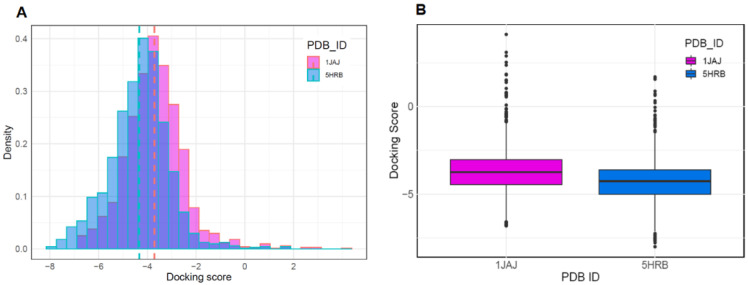
The DrugBank datasets were docked against the apo and DNA complex structure of *Asfv*PolX protein. The distributions of docking scores for each structure are shown as histogram (**A**) and box and whisker plots, with outliers indicated by black dots (**B**).

**Figure 3 molecules-26-03592-f003:**
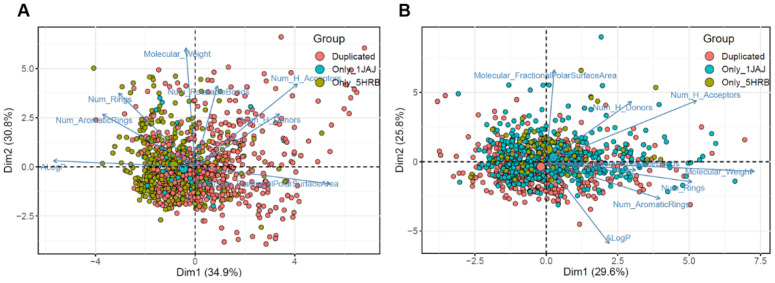
Visual representation of the chemical space of 1JAJ and 5HRB datasets. Distribution of the chemical space of the compounds in the 1JAJ and 5HRB datasets, according to principal component analysis. (**A**) Compounds with ≤−4.0 kcal/mol binding free energies. (**B**) Compounds with ≥−4.0 kcal/mol binding free energies. The loading plot vectors are represented by arrows for each physicochemical property.

**Figure 4 molecules-26-03592-f004:**
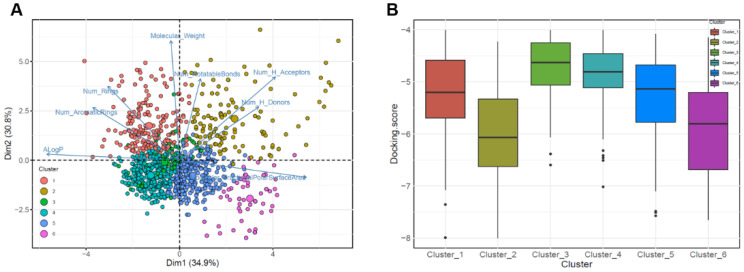
Principal component analysis clustering plot of the chemical space of prepared datasets. The loading plot vectors are represented by arrows for each physicochemical property. (**A**) The red, mustard, green, cyan, blue, and magenta dots correspond to clusters 1, 2, 3, 4, 5, and 6, respectively. Data points are color-coded by clusters of molecules. (**B**) Boxplots for SP Glide score distributions of prepared dataset for each cluster (Clusters 1–6).

**Figure 5 molecules-26-03592-f005:**
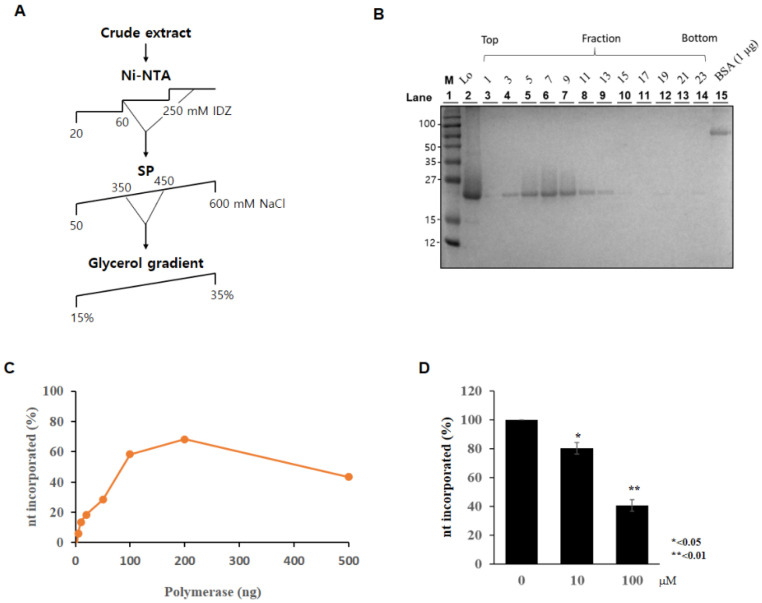
Pentagastrin inhibits *Asfv*PolX activity. (**A**) Schematic illustration of recombinant PolX purification. Ni-NTA, Ni^2+^-nitrilotriacetic acid affinity; SP, SP column chromatography; IDZ, imidazole. Glycerol gradient indicates the final purification procedure using glycerol gradient sedimentation. (**B**) Twelve fractions collected from glycerol gradient sedimentation were analyzed in a 15% SDS-PAGE and then the gel was stained with Coomassie Brilliant Blue R-250. Lo, Load. Molecular mass markers are indicated in kDa. (**C**) Polymerase activity was determined by measuring nt incorporation using indicated amounts of purified AsfvPolX protein. (**D**) Pentagastrin was mixed with 100 ng of ASFV PolX for 10 min and polymerase activities were measured as described in Materials and Methods.

**Figure 6 molecules-26-03592-f006:**
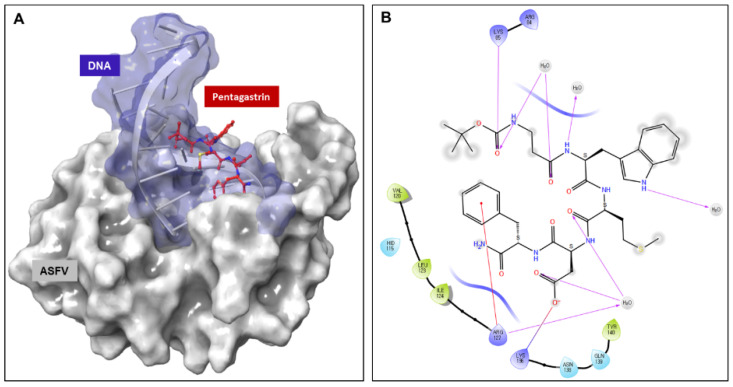
Molecular docking of pentagastrin to AsfvPolX protein. (**A**) Three-dimensional ligand interaction of pentagastrin generated by using Schrödinger software. The binding site in pentagastrin is shown as a surface model (**B**). Two-dimensional ligand interaction diagram of pentagastrin with the essential amino acid residues at the DNA binding site are represented in circles.

## Data Availability

Data sharing not applicable.

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
