# Peer review of "Prediction of African Swine Fever Virus Inhibitors by Molecular Docking-Driven Machine Learning Models"

_molecules, 2021, doi:10.3390/molecules26123592_

Round 1
Reviewer 1 Report
In this study, Choi et al. proposed a bioinformatic study to predict African Swine Fever Virus Inhibitors using Molecular Docking-Driven Machine Learning Models. Since the idea may be of interest, there are some major points that need to be addressed:
1. The authors should add more related works on bioinformatics in the literature review section. Currently, they did not show many bioinformatics-based works on African Swine Fever Virus Inhibitors.
2. Can the authors compare their results/findings with the previously published works?
3. The authors want to propose a method with the use of machine learning, but they did not describe clearly this machine learning part and how it plays an important role in molecular docking study.
4. More discussions should be added.
5. The authors should have some validation data.
6. Machine learning has been used in bioinformatics studies i.e., PMID: 33036150 and PMID: 33260643. Therefore, the authors are suggested to refer to more work in this part.
7. Source codes should be provided for replicating the work.
8. Quality of figures should be improved.
Round 2
Reviewer 1 Report
My previous comments have been addressed well.
Reviewer 2 Report
Excellent! I'm content with the author's response to my questions and changes to the paper. I believe there are many good clarification figures in the response that should be part of the supplementary information.